

# Emotional competence relating to perceived stress and burnout in Spanish teachers: a mediator model

Lourdes Rey[1], Natalio Extremera[2] and Mario Pena[3]

[1] Department of Personality Assessment and Psychological Treatment, Faculty of Psychology, University of Málaga, Málaga, Spain
[2] Department of Social Psychology, Faculty of Psychology, University of Málaga, Málaga, Spain
[3] Department of Educational Research and Diagnosis in Education, Universidad Nacional de Educación a Distancia (UNED), Spain

## ABSTRACT

This study examined direct associations between emotional competence, perceived stress and burnout in 489 Spanish teachers. In addition, a model in which perceived stress mediated pathways linking emotional competence to teacher burnout symptoms was also examined. Results showed that emotional competence and stress were significantly correlated with teacher burnout symptoms in the expected direction. Moreover, mediational analysis indicated that perceived stress partly mediated the relationship between emotional competence and the three dimensions of burnout even when controlling for salient background characteristics. These findings suggest an underlying process by which high emotional competence may increase the capacity to cope with symptoms of burnout, by reducing the experience of stress. Implications of these findings for future research and for working with teachers to prevent burnout are discussed.

# INTRODUCTION

Since the introduction of the concept of burnout by *Freudenberger* (*1974*), this syndrome is becoming an important topic in psychological and educational research (*Kyriacou*, *2001*; *Maslach, Schaufeli & Leiter*, *2001*). An expanding body of research examining the relationship between teaching stressful situations and burnout have found that teachers tend to report higher levels of occupational-related stress and burnout than many other human service professionals (*Maslach, Schaufeli & Leiter*, *2001*). According to *Maslach & Jackson* (*1986*), burnout syndrome is characterized by three primary dimensions: (1) feelings of emotional exhaustion and being emotionally drained by intense contact with students; (2) depersonalization or negative, cynical attitudes toward students; and (3) a sense of lack of personal accomplishment or of ineffectiveness and low competence in one's work. This syndrome has been repeatedly described as being the result of a significant accumulation of chronic work-related stress (*Maslach*, *2003*). In identifying important sources of teacher burnout, *Montgomery & Rupp* (*2005*) included a wide diversity of stressors such as students' misbehaviours and discipline problems, students' poor motivation for work, role conflict and role ambiguity, and pressure and criticisms from parents, among others. All these

Corresponding authors
Lourdes Rey, lrey@uma.es
Mario Pena, mpena@edu.uned.es

factors have been associated with the development and maintenance of teacher burnout (*Chang, 2013*; *Lambert & McCarthy, 2006*). In addition, the high levels of stress experienced by teachers and costly mental health-related problems have also been found to contribute to the growing retention problems within the school setting (*Sass, Seal & Martin, 2011*).

However, teaching professionals do show marked individual differences in their responses to different teaching stressors, ranging from minor psychological symptoms and mild anxiety to burnout as the more severe negative affective experience (*Bauer et al., 2006*). Individual differences constructs are believed to increase the likelihood that conditions related to the stressful situation will be effectively addressed or resolved and higher levels of such skills will lead to lower reported burnout (*Langelaan et al., 2006*). Consistent with this view, researchers in the field of individual differences have shown interest in examining mechanisms that might explain what individual factors predispose teachers to burnout (*Aloe et al., 2014*; *Chang, 2013*).

One dimension that has gained growing attention in the field of individual differences as a person-related psychological resource is the people ability to deal effectively with affective information, often referred to as emotional competence (or emotional intelligence) (*Mayer, Salovey & Caruso, 2004*; *Ciarrochi & Scott, 2006*). Some authors prefer the use of the more generic and neutral "emotional competence" term when are used self-report measures similar to those used in this study (*Ciarrochi & Scott, 2006*; *Giardini & Frese, 2006*). Emotional competence (EC) is regarded here as an affect-related skill, which refers to the ability to understand, manage, and express the social and emotional aspects of life in ways that enable one to coping with stressful and emotionally laden situations (*Kotsou et al., 2011*). In recent years, studies and theories have emphasized that EC has an impact on adaptive functioning at school (*Jennings & Greenberg, 2009*), even in the prediction of prosocial classsroom climate and teacher burnout (*Chan, 2006*; *Platsidou, 2010*).

Several authors have proposed that EC is a personal coping resource that allows teachers to manage and cope successfully with external and internal demands in a stressful workplace (*Brackett & Katulak, 2006*; *Jennings & Greenberg, 2009*). According to *Zeidner* (*2009*), it is believed that emotionally competent individuals are better adjusted than their less skilled counterparts due to the former having access to their own feelings and being able to manage their negative states even when emotionally aroused by challenging situations. Likewise, *Chan* (*2006*) suggested that the integrated operation of these emotional competencies might render teachers less vulnerable to teacher burnout, since they might gain better access to the healthy information and action tendency within emotions, and use the information to make sense of their reactions to stressors as well as to guide adaptive action. In fact, EC has been considered to be particularly relevant to work behaviour in teachers because abilities to manage relationships and to show interpersonal sensibility when needed are the prerequisites for improving increasing classroom management (*Garner, 2010*). Consistent with this assumption, accumulating empirical research has shown that EC might explain unique variance of both attitudinal and behavioural work outcomes (*Garner, Moses & Waajid, 2013*; *Giardini & Frese, 2006*). Having high EC is one of these personal resources that might act as a protector factor, reducing the effects of teacher work strain (*Brackett et al., 2010*; *Jennings, 2011*).

Although EC appears to be negatively related to burnout, the mechanisms by which EC affects burnout are not clear. *Zeidner* (*2009*) have listed several underlying mechanisms which might play a mediating role in the EC–occupational-related stress link, suggesting the levels of perceived stress as an underlying process. Given the emerging evidence of a link between EC and burnout (*Brackett et al.*, *2010*; *Jennings*, *2011*; *Platsidou*, *2010*), and the well-established connections between high perceived stress and burnout (*Maslach*, *2003*), it is probable that part of the influence of EC on burnout occurs through the mediating effect of perceived stress. However, although these factors are reported to be correlated with each other in the literature, such a proposition has not been empirically evaluated. Also, the relative strength of the direct and indirect relations between EC and burnout is unclear. Therefore, there are some reasons to believe that EC may show influence the risk of burnout symptoms through perceived stress. Stress is conceptualized as an experience resulting from an imbalance between demands and resources or as taking place when pressure exceeds one's perceived ability to cope (*Lazarus*, *2006*). Relevant to the current study, results of previous research suggest that teachers confused about their emotional knowledge abilities and with no confidence in their own emotional regulation abilities are unlikely to feel in control of stressful situations, showing generally higher perceived stress (*Brackett et al.*, *2010*; *Jennings*, *2011*). When these individuals maintain higher stress levels, there is the potential for them to increase the risk of becoming burned out on the job. On the contrary, it is tentative to think that teachers who are high on emotional competence would also experience as though they have more control over their stressful environment because they can confer sense and manage their negative moods more adaptively (*Zeidner*, *2009*). In fact, several researchers have found evidence that EC are negatively associated with experience of stress (*Collie, Shapka & Perry*, *2012*; *Garner*, *2010*), supporting its utility as an individual characteristic that helps shape subjective interpretations of stressful daily events. These positive effects of EC on perceived stress may result in reduced levels of burnout. Likewise, EC might influence the manner in which a teacher views and reacts to stressful events and experience of stress which, in turn, may reduce levels of burnout symptoms. Hence, the assessment of perceived stress as a potential mediator of the link between EC and teacher burnout symptoms seems to be warranted.

Additionally, a variety of socio-demographic factors has been associated with symptoms of burnout, including age, sex and grade level taught (*Maslach, Schaufeli & Leiter*, *2001*). In order to ensure that our findings would not be confounded with these predictive dimensions, respondents's age, gender, and grade level taught were used as control variables in our proposed mediation model of perceived stress on the link between EC and burnout.

In summary, a large body of literature deals with the prediction of different indicators of teacher burnout. One of this predictor may be the way in which people identify and manage one's own feelings and those of others. However, rather little is known about the mechanisms that interlink EC, perceived stress and burnout in teachers. Beyond the mere associations among these dimensions, this study will examine whether there is a mediation effect to gain more insight into possible mechanisms of the development of teacher burnout and of protective resource factors. Although the theoretical assumption of mediation of perceived stress on the interplay between EC and burnout has been proposed

by authors (*Zeidner*, *2009*), one finds hardly any formal tests of mediation in the available literature. One valuable exception is the preliminary work of *Durán, Extremera & Rey* (*2009*) who examined the relationship of perceived emotional intelligence, stress, and burnout in a sample of high school teachers using the TMMS (i.e., Trait Meta-Mood Scale) to predict teacher burnout and the mediating role of stress. In short, they found partial support for the idea that perceived stress mediated the influence of TMMS dimensions on specific subscales of teacher burnout. However, as authors have pointed out, TMMS only evaluates intrapersonal emotional process, but this does not preclude the importance of interpersonal dimensions not measured by TMMS. Besides, other important socio-demographic dimensions such as sex, age and grade taught level, which are related to stress and teacher burnout, were not controlled for. Finally, the sample was composed of only high school teachers which limits the extent to which the findings can be generalized. Future research with a heterogenous sample of grade level teachers, using more comprenhensive EI measures and controlling for classic socio-demographics factors related to teacher burnout is needed to confirm the usefulness and validity of these findings. In light of the above considerations, the purpose of the present study was to examine the role of perceived stress as a mediator in the relationship between EC and teacher burnout. We hypothesized that more emotionally competent teachers would experience less stress, and this reduction in stress might account for their decreased experience of symptoms of teacher burnout, even when controlling for salient background characteristics.

## METHODS

### Participants

The study used a cross-sectional survey with incidental sampling. The original sample, who participated voluntarily and anonymously in the study, consisted of 494 teachers (334 female and 160 male). The responses provided by five participants were dropped from the study because they were incomplete. The final sample comprised 489 teachers (330 females and 159 males) from fifteen autonomous regions in Spain: Andalusia ($n = 165$); Galicia ($n = 80$); Madrid ($n = 71$); Castile-La Mancha ($n = 46$); Castile and León ($n = 11$); Cantabria ($n = 25$); Valencia ($n = 17$); Navarra ($n = 15$); Extremadura ($n = 11$); Aragón ($n = 11$); Canary Islands ($n = 9$); Asturias ($n = 9$); Balearic Islands ($n = 8$); Ceuta ($n = 6$) and La Rioja ($n = 5$). The mean age was of 39.90 years (SD = 9.49 years). One hundred and three were elementary teachers (males = 14; females = 89) who taught pupils from 3-to-5 year-olds; 243 intermediate teachers (males = 77; females = 166) who taught students from 6-to-11 year-olds; 135 secondary teachers (males = 66; females = 69) who taught students from 12-to-17 year-olds and eight teachers with unreported grade level taught. A total of 81.8% of the teachers worked in state schools and 17.2% worked in private schools that receives public funds. Teaching experience ranged from one month to 38 years ($M = 4.22$ years; SD = 7.85 years).

## Materials

### Wong and Law Emotional Intelligence Scale (WLEIS; *Law, Wong & Song, 2004*)

To evaluate EC we used the WLEIS. The scale consists of 16 items relating to self-emotional appraisal (SEA), other emotional appraisal (OEA), use of emotion (UOE) and regulation of emotion (ROE). The items are scored on a 7-point Likert-type scale (1 = totally disagree to 7 = totally agree). The scale has shown adequate internal consistency and evidence of validity (*Law, Wong & Song*, *2004*). As in previous studies, we combined the subscales into a global EC measure since we were more interested in the overall EC construct than in each of its dimensions (*Sy, Tram & O'Hara*, *2006*; *Wong, Wong & Peng*, *2010*). The WLEIS was translated from English into Spanish using the method of back-translation. In our sample, the Cronbach's alpha for total EC was 0.89.

### Perceived Stress Scale (PSS; *Cohen, Kamarck & Mermelstein, 1983*)

This is a 14-item measure of self-appraised stress. Respondents are asked to rate the frequency during the last month with which they have been in situations they consider stressful. Frequency is rated across a 5-point Likert-type scale ranging from 0 (never) to 4 (very often). We used the Spanish version of this scale (*Remor & Carrobles*, *2001*). In our sample, the reliability was 0.75.

### Maslach Burnout Inventory (MBI; *Maslach & Jackson, 1986*)

This scale consists of 22 items scored on a 7-point frequency scale from 0 (never) to 6 (daily) and comprises three sub-scales: Emotional Exhaustion, which describes feelings of being emotionally overextended and exhausted by one's work; Depersonalization, which describes an unfeeling and impersonal response towards recipients of one's care or service; and Personal Accomplishment, which describes feelings of competence and successful achievement in one's work with people (*Maslach & Jackson*, *1986*). We used the most widely used Spanish version of MBI, which has shown adequate psychometric properties in occupational population (*Seisdedos*, *1997*). In this study, Cronbach's alpha for Emotional Exhaustation was 0.90; for Depersonalizaton was 0.72 and for Personal Accomplishment was 0.83.

## Procedures

Participants were recruited through the help of Psychopedagogy students (UNED) and psychology students at University of Málaga. These students received instructions from the teaching staff regarding how to administer the questionnaire correctly. The data were collected as part of a project, participating in a research study on emotions and well-being at work. Most importantly, participants were fully informed about the voluntary basis of participation; they were also informed that questionnaires were completed anonymously. Thereafter, they provided informed consent. Once the questionnaires were completed, the students returned them to the teaching staff for statistical processing. The study protocol was approved as part of the project PSI2012-38813 by the Research Ethics Committee of the University of Malaga.

**Table 1  Descriptives, internal reliabilities and correlations for all study measures.**

|  | M | SD | Skewness | Kurtosis | α | 1 | 2 | 3 | 4 | 5 |
|---|---|---|---|---|---|---|---|---|---|---|
| 1. Emotional competence | 5.25 | .69 | −0.54 | 0.15 | .89 | – | | | | |
| 2. Perceived stress | 1.19 | .64 | 0.41 | 0.08 | .75 | −.47[**] | – | | | |
| 3. Emotional exhaustion | 2.06 | 1.24 | 0.56 | −0.25 | .90 | −.38[**] | .45[**] | – | | |
| 4. Depersonalization | .84 | 1.02 | 1.53 | 2.48 | .72 | −.32[**] | .27[**] | .52[**] | – | |
| 5. Personal accomplishment | 4.78 | .84 | −.99 | .96 | .83 | .55[**] | −.34[*] | −.42[**] | −.40[**] | – |

Notes.
[**]$p < 0.01$.
[*]$p < 0.05$.

# RESULTS

## Descriptive statistics

Descriptive statistics (e.g., means, standard deviations, skewness and kurtosis) alpha coefficients for the study variables are presented in Table 1.

Mahalanobis' distance was computed to determine if multivariate outliers occurred in the dataset. One participant was found to be outlier ($p < .001$), being removed from the dataset. Pearson product–moment correlation coefficients were used to specify the relationship between the variables. Scores on all the factors, except for depersonalization, are normally distributed, as indicated by skewness and kurtosis. Therefore, for depersonalization that showed high kurtosis, Spearman correlations were computed. In general, all relationships among EC, stress and burnout symptoms were statistically significant. Thus, perceived stress scores were significantly and positively related to emotional exhaustation and depersonalization and negatively associated with personal accomplishment.

## Mediation analysis

According to the general guidelines offered by *Baron & Kenny* (*1986*) for establishing evidence for a proposed mediation model, four conditions need to be satisfied. First, the predictor (EC) needs to be related to the outcome (burnout subscales). Second, the predictor must be related to the hypothesized mediator (perceived stress). Third, the mediator must be related to the outcome when controlling for the predictor. Fourth, the effect of the predictor on the outcome must be reduced when the effect of the mediator is controlled. As recommended by *Preacher & Hayes* (*2008*), bootstrap analysis, a non-parametric sampling procedure, was used to test the significance of the indirect effects. Bootstrapping is an approach that resamples the original sample size from the data multiple times and does not rely on the assumption that data are normally distributed. Current analyses utilized 5,000 bootstrap resamples to generate 95% confidence intervals. The indirect effect is significant at $p < .05$ if zero is not included in the 95% confidence interval for that indirect effect. We conducted a set of path analyses to test separately the potential role of perceived stress as a mediator between EC and each burnout subscale including sex, age, and grade taught level as covariates in the analyses.

**Table 2  Regression models predicting the three teacher burnout symptoms from emotional competence and perceived stress controlling for sex, age and grade taught level.**

| Predictor | Emotional exhaustion | | | | Depersonalization | | | | Personal accomplishment | | | |
|---|---|---|---|---|---|---|---|---|---|---|---|---|
| | $R^2$ | $F$ | $\beta$ | $\Delta R^2$ | $R^2$ | $F$ | $\beta$ | $\Delta R^2$ | $R^2$ | $F$ | $\beta$ | $\Delta R^2$ |
| **Step 1** | .04 | 6.23 | | | .03 | 4.30 | | | .04 | 5.61 | | |
| Sex | | | .06 | | | | −.10[*] | | | | −.00 | |
| Age | | | .19[**] | | | | .04 | | | | −.08 | |
| Grade taught level | | | .02 | | | | .09 | | | | −.16[**] | |
| **Step 2** | .18 | 24.42 | | .14[**] | .13 | 17.45 | | .10[**] | .35 | 60.89 | | .31[**] |
| Sex | | | .05 | | | | −.10[**] | | | | −.00 | |
| Age | | | .16[**] | | | | .02 | | | | −.04 | |
| Grade taught level | | | .00 | | | | .07 | | | | −.13[**] | |
| EC | | | −.37[**] | | | | −.33[**] | | | | .56[**] | |
| **Step 3** | .27 | 32.89 | | .09[**] | .15 | 16.47 | | .02[**] | .36 | 51.12 | | .01[**] |
| Sex | | | .03 | | | | −.11[**] | | | | .01 | |
| Age | | | .14[**] | | | | .01 | | | | −.03 | |
| Grade taught level | | | −.00 | | | | .07 | | | | −.13[**] | |
| EC | | | −.22[**] | | | | −.25[**] | | | | .50[**] | |
| Perceived stress | | | .34[**] | | | | .16[**] | | | | −.12[**] | |

**Notes.**
[*] $p < .05$.
[**] $p < .01$.

## Perceived stress as a mediator between EC and teacher emotional exhaustation

We conducted a series of regressions to test for a mediation effect of perceived stress on the relationship between EC and teacher emotional exhaustion controlling for sex, age and grade taught level in the first step. As seen in Table 2, after controlling for teachers background characteristics, in the second step EC was regressed on emotional exhaustion, accounting for 14% of the variance in this burnout dimension. In the third step, emotional exhaustation was simultaneously regressed on background characteristics, EC and perceived stress. When perceived stress was entered into the equation, the beta weight from EC dropped from −.37 to −.22, but it remained significant.

By using the SPSS macro provided by *Preacher & Hayes* (*2008*), the indirect effect of perceived stress was estimated to lie between −.38 and −.19 with 95% confidence (bootstrap coefficient $= -.27$), as you can see in Table 3. Because zero was not in the 95% confidence interval, the findings suggested that the indirect effect through perceived stress was significantly different from zero at $p < .05$, supporting the view that perceived stress partly mediated the relation between EC and emotional exhaustation controlling for sex, age and grade taught level.

## Perceived stress as a mediator between EC and depersonalization

Next, EC was regressed on depersonalization ($b = -.32, p < .001$), accounting for 10% of the variance in this burnout dimension even controlling for background characteristics. In the third step, depersonalization was simultaneously regressed on background

**Table 3  Indirect effects of emotional competence on burnout symptoms, through perceived stress and controlling for background characteristics.**

| Model[a] | Estimated[b] coefficient | SE | 95% CI | |
|---|---|---|---|---|
| | | | Lower | Upper |
| Emotional competence-emotional exhaustation | −.29 | .04 | −.38 | −.19 |
| Emotional competence-depersonalization | −.11 | .03 | −.19 | −.05 |
| Emotional competence-personal accomplishment | .06 | .02 | .02 | .12 |

Notes.
[a] In the presence of perceived stress as a mediator, and the covariates of, sex, age, and grade taught level.
[b] Estimated using bias corrected and accelerated bootstrapping, with 5,000 samples.

characteristics, EC and perceived stress. As Table 2 shows, when perceived stress was included in the model, there was a decrease in EC influence on depersonalization (the beta weight dropped from −.33 to −.25). The indirect effect was significant (bootstrap coefficient = −.11), with a 95% confidence interval of −.18 to −.04, supporting the view that perceived stress partly mediated the relation between EC and depersonalization (see Table 3).

## Perceived stress as a mediator between EC and personal accomplishment

Finally, we conducted a series of regressions to test for a mediation effect of perceived stress on the relationship between EC and personal accomplishment. In the first step, we controlling for background characteristic, grade taught level was a significant predictor of the variance in this burnout dimension. In the second step, EC was found to be a significant predictor of personal accomplishment, accounting for 31% of the variance in this burnout symptom. In the last step, perceived stress was added to the model, explaining an additional 1% of the variance in this burnout subscale. As Table 2 shows, when perceived stress was included in the model, there was a decrease in EC influence on personal accomplishment (the beta weight dropped from .56 to .50). The indirect effect was significant (bootstrap coefficient = .06), with a 95% confidence interval of .02 to 0.11, supporting the view that perceived stress is also a partial mediator in the link between EC and personal accomplishment (see Table 3).

## DISCUSSION

This study expands on previous research literature on EC and teacher burnout (*Chan*, *2006*; *Jennings*, *2011*; *Platsidou*, *2010*) by examining direct and indirect relations between EC, perceived stress and burnout in a sample of teachers. Our underlying reasoning was that EC may work through to the reduction of perceived stress, which in turn would promote lower levels of teacher burnout independently of sex, age and grade taught level.

There is accumulating empirical evidence that deficits in emotional abilities are directly related to high levels of burnout (*Durán, Extremera & Rey*, *2009*; *Platsidou*, *2010*). Consistent with these findings, our research did provide evidence that EC was correlated with both perceived stress and burnout symptoms. Specifically, teachers who report being poor emotionally at perceiving and managing emotions also tend to report feeling exhausted more often and being more cynical about their work and about students whom

they work with, and a more reduced sense of personal accomplishment. One explanation for these findings is that teaching professionals' skills at effectively managing emotional challenges are characterized by more constructive thought patterns, and teachers find it easier to catch and to identify faulty appraisals and correct maladapted construals in the workplace (*Zeidner*, *2009*). In addition, it is possible that teachers who are more emotionally competent would also feel that they have more control over their stressful environment because they can confer sense and manage their negative moods associated with stress more adaptively.

Similarly, in line with earlier studies, our correlational data found that greater perceived stress scores were related in the expected direction to higher levels of burnout symptoms (*Maslach*, *2003*). Specifically, greater perceived stress was significantly associated with greater emotional exhaustion and depersonalization and lower feelings of personal accomplishment. The present study found some preliminary support for the idea that perceived stress mediated the influence of EC on specific subscales of teacher burnout, suggesting a complex process. The mediation analyses revealed that perceived stress partially mediated the relations between EC and all burnout symptoms. The results of this study might point to one plausible explanation for the association between EC and burnout as being related to the reduced perceptions of stress experienced by teachers with high EC. In general, increasing EC in teachers may reduce the risk of job burnout both directly and indirectly by reducing the experience of stress. Emotionally competent teachers may reduce be better equipped to engage with and successfully manage stressful school situations, which in turn would lead to lower symptoms of emotional exhaustation and depersonalization and higher personal accomplishment. More research is needed to examine the underlying mechanism by which teachers with high EC reduce their levels of stress, either identifying and avoiding potentially stressful contexts or interpreting stressful conditions at work in a less stressful way.

Nevertheless, these findings may be valuable not only for developing theoretical models and for understanding better the link between EC and occupational stress (*Zeidner*, *2009*), but also for developing more effective stress intervention programmes for teachers (*Chang*, *2013*). Based on our results, it is suggested that EC might play an important role in the process of certain responses to job stress in teachers. In terms of intervention implications, teachers could be taught how to employ specific emotional strategies for managing stress; this in turn might help them to reduce their feelings related to teacher burnout. In this sense, our data support the idea that programmes to prevent burnout involving the enhancement of EC might be useful to facilitate an increased perception of self-appraisal in teaching work. Besides, whereas therapeutic efforts to increase EC may reduce symptoms burnout, perceived stress appears to be an important aspect of burnout in teachers, so it is also of great importance to help teachers who are vulnerable for burnout to provide specific strategies for reducing negative moods provoked by stress. Cognitive behavioral therapy has been shown to be an effective way of reducing work-related stress, even more than the other intervention types (*Van der Klink et al.*, *2001*). Further research will determine whether EC interventions, either alone or in conjunction with other stress management interventions such as cognitive behavioral therapies, will enhance psychological outcomes

for the classic symptoms of teacher burnout (*Farber*, *2000*). Future work should also be addressed to the positive pole of burnout, work engagement (*Bakker*, *2011*), to determine when and under what circumstances EC work better for increasing individuals' positive and fulfilling attitude toward work in teachers.

Finally, several limitations of the current study should be mentioned. First, although our data provide preliminary evidence for the mediation model proposed, due to data being collected in a single wave of measurement it is impossible to determine the directionality of any causal relationships between variables. A closer examination of the nature of this relationship should be performed using prospective studies. Second, self-reported measures of EC may not reflect actual emotional ability. To address this issue, further studies might consider the inclusion of other approaches complementary to the use of self-report for measuring EC, such as peer reports or performance measures, which provide valuable information about people's emotional reactions in the work sphere. Finally, it is important to underline that our participants were based on a purposive sampling and, hence, their representativeness is questionable.

Despite these limitations, it appears that EC may have an important place in the educational psychologist's repertoire of occupational well-being and may suggest possible avenues for improving future stress management programs to assist distressed teachers. Although school climate factors are considered to be one of the prominent causes of teacher burnout (*Aloe et al.*, *2014*; *Grayson & Alvarez*, *2008*), our results, in line with prior studies, support the importance of individual differences with regard to EC in the development of teacher burnout. While organization-focused interventions in educational settings are important in preventing burnout, the findings of this study provide some evidences that educational settings should pay more attention to the development of protective personal resources taking into account the perspective of interaction between personal characteristics and contextual variables in teaching (*Aloe et al.*, *2014*; *Chang*, *2013*).

Finally, as high stress levels and poor EC continually rank as the most prominent causes why teachers become dissatisfied with the profession and end up leaving their positions (*Chang*, *2009*), paying more attention to the development of these personal resources might help teachers employ EC in their professional and personal relationships and contribute to greater positivity at work (*Garner*, *2010*; *Jennings*, *2011*). Future research could examine this issue by training people in EC and observing how such training impacts on teacher burnout symptoms over time. Elucidating which protective resource factors contribute most to a better subjective well-being at work, and how to develop these aspects, is a promising approach to the enhancement of teaching skills and merits serious attention.

## ACKNOWLEDGEMENTS

We would like to give our heartfelt thanks to all the teachers who participated in this study.

### Funding

This research was supported by the Spanish Ministry of Economy and Competitiveness (Grant PSI2012-38813). The funders had no role in study design, data collection and analysis, decision to publish, or preparation of the manuscript.

### Grant Disclosures

The following grant information was disclosed by the authors:
Spanish Ministry of Economy and Competitiveness: PSI2012-38813.

### Competing Interests

The authors declare there are no competing interests.

### Author Contributions

- Lourdes Rey, Natalio Extremera and Mario Pena conceived and designed the experiments, performed the experiments, analyzed the data, contributed reagents/materials/analysis tools, wrote the paper, prepared figures and/or tables, reviewed drafts of the paper.

### Human Ethics

The following information was supplied relating to ethical approvals (i.e., approving body and any reference numbers):
Comité ético de experimentación de la Universidad de Málaga.

### Data Availability

The raw data has been supplied as Supplemental Information.

### Supplemental Information

Supplemental information for this article can be found online at http://dx.doi.org/10.7717/peerj.2087#supplemental-information.

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
