# Peer review of "Emotional competence relating to perceived stress and burnout in Spanish teachers: a mediator model"

_PeerJ, doi:10.7717/peerj.2087_

## Round 0.1 · original submission · Major Revisions

I believe reviewer 2 has challenged the conceptual validity of this study. Therefore, I'd suggest the authors not only revise or refute their manuscript according to reviewer 1's suggestion, but also address the comment of reviewer 2's challenge by explaining the rationale of examining the mediating role of perceived stress.

Reviewer 1 ·

Basic reporting

The submitted paper entitled “Emotional competence relating to perceived stress and burnout in Spanish teachers: a mediator model” seeks to examine the relationship between emotional competence, perceived stress, and burnout in Spanish teachers; and examining the mediating effects of perceived stress on the emotional competence and burnout. The authors had clearly examined the background of the question so to make this paper worthy of being published on PeerJ. However, after reading the whole manuscript, there are several places need to be modified to make this paper better as follow:

Experimental design

#1, line 157, should be “participants and procedures” instead of “participants and procedure”

#2, line 158, you said “…The study was a cross-sectional questionnaire design…” When you said “cross-sectional questionnaire design” it seems that you said two things together --- “research design” and “questionnaire survey.” Why not just put it straight forward said you used a “cross-sectional survey” to do your study.

#3, line 161-169, I found your calculation about the participants were wrong. You said you totally had 489 valid participants. But I add 103 (elementary teachers) +243 (intermediate teachers) +132 (secondary teachers) = 478.

#4, line 175, I suppose you used the scale of Law, Wong, & Song (2004) instead of Wong & Law (2002) because Wong & Law (2002) was not a paper for scale development and validaiton.

#5, line 203-208. Did you analyze the statistical properties of all the investigated variables including skewness, kurtosis, and outlier. Skipped these basic examinations before main analyses may cause systematic errors.

Validity of the findings

#6, line 321-331, I would cautiously explain the role of emotional competence on coping skills and stress. Past emotional competence theory suggest that high in EC implies also being good in stress management skills (e.g., Bar-On, 2002). But it should not explain that ….high in EC may be also effective to focus on providing coping skills ….(line 322-323). This is just over speculated.

Additional comments

I consider this is a well-prepared paper worthy of being published on PeerJ. However, I would like to see authors make some more revisions and changes so to make this paper better.

Reviewer 2 ·

Basic reporting

No Comments

Experimental design

No Comments

Validity of the findings

No Comments

Additional comments

The main purpose of this study was to examine whether perceived stress was a mediator of the relationship between emotional competence and burnout. The authors argued that testing this mediator model would help to understand the mechanism by which emotional competence affected burnout. However, this model is unfortunately a tautology. Emotional competence is the ability to cope with stress (individuals with higher emotional competence have lower perceived stress) and burnout is the consequence of stress (individuals with higher perceived stress have severe burnout symptoms). Perceived stress is actually part of the concepts of both emotional competence and burnout. It should not be a third variable that will be considered as the mechanism or a possible mediator. Examining whether perceived stress is a mediator of the relationship between emotional competence and burnout is illogical and provide very little information to the understanding of either emotional competence or burnout.

---

## Round 0.2 · Minor Revisions

Both reviewers are generally satisfied with the revised manuscript and the authors have my congratulations. However, some minor issues need to be addressed before I can accept your manuscript. Specifically the authors are recommended to describe how they conducted the data collection, such as via e-mail, mail, or in person, as well as a brief statement of research procedure. In addition, the citations in the introduction and discussion sections were relatively old. Please update.

Reviewer 1 ·

Basic reporting

The submitted paper entitled “Emotional competence relating to perceived stress and burnout in Spanish teachers: a mediator model” examined the relationship between emotional competence, perceived stress, and burnout in Spanish teachers; and the mediating effects of perceived stress on the emotional competence and burnout. The authors had revised the mistakes/errors that I have commented on previous manuscript. So I recommend it to be published on PeerJ.

Experimental design

reasonable and appropriate

Validity of the findings

The results can contribute to extant literature. Especially the results not only explain why high emotional intelligence perceived lowered burnout ( the first generation question) but also tell us which factor mediates this relationship (second generation question).

Additional comments

no

Reviewer 3 ·

Basic reporting

1) Clear, unambiguous, professional English language used throughout.
Overall this study is clear and free of any grammatical or structural written errors.

2) Intro & background to show context.
The paper clearly introduces the relationships among teacher burnout, teaching stressors, and emotional competence. Based on the previous findings, the authors have appropriately evidenced that perceived stress might be a potential mediator of the link between emotional competence and teacher burnout.

3) Literature well referenced & relevant.
Literature has been well-referenced and relevant. The format follows the standard of APA style.

4) Structure conforms to PeerJ standard, discipline norm, or improved for clarity.
The structure of the manuscript conforms to the standard of the PeerJ and the norm of relevant disciplines.

5) Figures are relevant, high quality, well-labelled & described.
Figures and tables are relevant and well-labelled that help readers realize the relationships among the variables, as well as the results of regression and mediation analyses.

Experimental design

1) Original primary research within Scope of the journal.
This study is well-focused and properly builds on previous work. The purpose of this study falls within the scope of the journal, which is to examine the mediating effect of perceived stress on the association between emotional competence and teacher burnout.

2) Research question well defined, relevant & meaningful. It is stated how research fills an identified knowledge gap.
The description of research question is well-stated that makes the potential of this study to bring new relevant information to the field clear at this point. Based on the previous findings, the authors have criticized the existing knowledge gaps and then pointed out how this study can solve these questions.

3) Rigorous investigation performed to a high technical & ethical standard.
This study is well designed and also follows the ethical standard to claim the approval of the IRB while recruiting teachers as the participants. However, the authors are recommended to describe how they conducted the data collection, such as via e-mail, mail, or in person, as well as a brief statement of research procedure.

4) Methods described with sufficient detail & information to replicate.
The materials and methods described in the manuscript are moderately sufficient for other researchers to replicate.

Validity of the findings

1) Negative/inconclusive results accepted.
In general, all relationships among emotional competence, stress and burnout symptoms were statistically significant, which were mostly consistent with the previous findings and the directions of the hypotheses.

2) Data is robust, statistically sound, & controlled.
The data provided is robust enough because this regression analyses could control the influences of differences in age, sex, and grade taught level of the participants, and then precisely indicated the prediction effects of emotional competence and perceived stress on teacher burnout. Also, this study utilized a robust method to examine the indirect effects of emotional competence on burnout symptoms through perceived stress.

4) Conclusion well stated, linked to original research question & limited to supporting results.
The conclusion is appropriately stated based on the results of this study and the findings of previous work. Also, the authors have raised their arguments and recommendations according to the findings of this study. However, the literature cited in the part of conclusion and introduction was somehow old. The authors are suggested to update the references cited in this manuscript.

Additional comments

Generally, the justification of research questions is clear which makes this study contribute to the field regarding teacher burnout. More, the method of this study is suitable and the validity and reliability of the measures are appropriate, which can increase the contribution of the findings. Therefore, this study is able to provide new information regarding the mechanisms of teacher burnout.

---

## Round 0.3 · accepted · Accept

The authors have addressed the comments to a satisfactory level such that now the manuscript can be accepted at its current condition. Congratulations.